# Se-O Bond Is Unique to High Se Enriched Sweet Potato Stem Protein with Better Antioxidant Ability

**DOI:** 10.3390/foods10123064

**Published:** 2021-12-09

**Authors:** Qi Gao, Jia-Le Wu, Lan-Ping Jiang, Su-Qi Sun, Xue-Jun Gu, Mei Tie, Masaru Tanokura, You-Lin Xue

**Affiliations:** 1College of Light Industry, Liaoning University, Shenyang 110036, China; gaoqi0925@163.com (Q.G.); alexander_wjl@163.com (J.-L.W.); lanping.jiang@link.cuhk.edu.hk (L.-P.J.); 15566030117@163.com (S.-Q.S.); 2Department of Regional Economic Development, Party School of Liaoning Provincial Party Committee, Shenyang 110161, China; 3Institute of Rare and Scattered Elements, Liaoning University, Shenyang 110036, China; sheshen311@126.com; 4College of Environment, Liaoning University, Shenyang 110036, China; tiemei@lnu.edu.cn; 5Department of Applied Biological Chemistry, Graduate School of Agricultural and Life Sciences, The University of Tokyo, Tokyo 113-8657, Japan

**Keywords:** Se, sweet potato stem protein, structure analyses, Se species, antioxidant activity

## Abstract

Sweet potato plants were treated with selenium (Se). Spraying Se on the sweet potato leaves was an effective Se enrichment method and proteins were extracted from the sweet potato stem. The structural characteristics of the protein were investigated. Fourier transform infrared spectroscopy (FT-IR) detected more signals from the Se-enriched sweet potato stem protein (SSP), and the number of forms of Se chemical bonds gradually increased with increasing Se content, such as the Se-O bond in high Se-enriched SSP, indicating altered secondary structures.Scanning electron microscopy-energy dispersive spectrometry (SEM-EDS) indicated more Se atoms in the Se-enriched SSPs (SSSPs). The DSC results revealed that Se enrichment enhanced the thermal stability of the samples. Moreover, selenomethionine (SeMet), selenocystine (SeCys_2_), and methylselenocysteine (MeSeCys) were determined to be the main Se forms in the SSSPs. Furthermore, the SSSPs showed relatively higher superoxide anion radical and DPPH radical scavenging activities than the blank, which indicates that SSSPs can be used as antioxidants. By recovering the proteins, the agricultural by-product—sweet potato stem can be further utilized, and the obtained Se-enriched proteins may contribute to human health.

## 1. Introduction

Selenium (Se), a trace mineral and an essential microelement that is of vital importance, is well known for its particular effects on gene expression and energy metabolism in living organisms as a component of the active sites of certain reductases and proteins and the World Health Organization (WHO) recommended daily intake levels range between 50–60 μg per day, with small differences based on the gender and the age groups [1].However, the distribution of Se is extremely uneven around the world; low-Se areas are common, and approximately 112 million people lack dietary Se [2]. Moreover, some diseases, such as bone loss and cardiovascular diseases, can be caused by dietary Se deficiency [3]. Daily food intake is the main source of Se for individuals, but the Se content in common foods does not currently meet the daily dietary nutrient requirements for humans. To increase dietary intake of Se, studies of Se-enriched foods, such as wheat, rice, broccoli, onion, pumpkin, cabbage, soybean, garlic and mushroom have been conducted [4,5].

However, there is evidence that inorganic Se species are toxic to animals, unlike organic Se species; the former may induce toxic effects in neurons that may be involved in amyotrophic lateral sclerosis etiology [6]. An effective way to increase Se levels is to consume Se-enriched plants because the organic forms of Se have higher bioavailability than the inorganic forms [7]. There are many available and effective ways to increase the Se content in plants, such as by spraying the plant with Se solution or by adding selenate or selenite to the soil [8]. However, high selenate or selenite concentrations can cause Se toxicity in plants. Notably, appropriately increasing the sprayed concentration of Se can slow fruit softening and enhance the health-related properties of some plants, such as pear and peach [9].

Sweet potato (*Ipomoea batatas* L.), which originates from Central America, is extensively cultivated in Asian, African and Latin American countries. China is the leading producer of sweet potatoes, accounting for approximately 76% of the total world production in 2011 [10]. Apart from the tubers, which are commonly consumed, the leaves of sweet potatoes are also consumed as leafy vegetables in Asian and African countries as a source of protein, dietary fiber, antioxidants, and minerals [11]. However, after the tuber harvest, most of the sweet potato leaves are wilted, and a large number of stems are left, which are used in feeding livestock or directly discarded. Therefore, the utilization of sweet potato stems, which contain approximately 11–14% crude protein, is of great interest [12].

The objective of this study was to investigate the best Se enrichment method for sweet potato stems, to characterize sweet potato stem protein (SSP), to identify the main Se species in Se-enriched SSPs and to characterize the in vitro antioxidant activity of SSPs with different levels of Se. By using the proteins effectively, the agricultural by-product—sweet potato stem can be further utilized, and the Se-enriched proteins may become a novel and valuable source contributing to human health.

## 2. Materials and Methods

### 2.1. Preparation of Se-Enriched Sweet Potato Stem

Se-enriched sweet potato plants were grown in Shenyang, Liaoning Province, China. The experimental field was divided into two areas. The Se application method in one area was spraying selenite solutions at different concentrations on the soil, and the method in the other area was spraying selenite solutions at different concentrations on the sweet potato leaves. The concentrations of sodium selenite (inorganic Se) applied ranged from 0 to 10.0 mg/mL. As the sweet potato leaves turned yellow and wilted during harvest, only the stems were harvested. The fresh Se-enriched sweet potato stems were washed with tap water, drained and dried in the sun. Then, the dried stems were ground and kept in a desiccator at room temperature until use.

### 2.2. Extraction of SSP

SSP was prepared using the alkaline extraction and acid precipitation method according to a previously described method [13]; 20 g of sweet potato stem powder was mixed with 800 mL of 0.05% sodium hydrogen sulfite solution (40:1 solvent-sample ratio, v/w; fridge-stored at 4 ℃), and the pH of the solution was adjusted to 10.0. After ultrasonic extraction for 25 min at 240 W by a TH-400 BQ CNC ultrasonic cleaner (Jining Tianhua Ultrasonic Electronic Instrument Co., Ltd., Jining, China), the solution was centrifuged at 6000 rpm for 30 min by a TG-16 centrifuge (Xiangtan Xiangyi Instrument Co., Ltd., Hunan, China). The pH of the supernatant was adjusted to 4.0 after filtration. Afterward, the mixture was centrifuged at 6000 rpm for 30 min. The precipitate, which was separated and collected in a beaker, was redissolved in distilled water, and 1.0 mol/L NaOH was added to the solution to adjust the pH to 7. The SSP solution was lyophilized (Approximately 0.6 g SSP powder was obtained from 20 g sweet potato stem powder) and stored in a desiccator until use.

### 2.3. Determination of the Total Se Contents in Stem Powder and SSP

Amounts of 0.25 g of stem powder and 0.1 g of SSP powder were weighed and subjected to microwave digestion. The cooled digestion solution was mixed with 5 mL of hydrochloric acid (6 mol/L) and heated to 150 °C. When the remaining solution was 2–3 mL, the heating was stopped, and the cooled solution was diluted up to 25 mL. The Se content was determined using an AFS-9800 atomic fluorescence spectrometer (Kechuang Haiguang Instrument Co., Ltd., Beijing, China) [13].

### 2.4. Sodium Dodecyl Sulfate-Polyacrylamide Gel Electrophoresis (SDS-PAGE) of SSP

According to a previously described method [13,14], SDS-PAGE of the SSP samples (SSP, low-Se SSP and high-Se SSP) was performed. About 50 μg SSP powder was dissolved in 2× sample buffer containing β-mercaptoethanol (β-ME). The sample solution of the protein was denatured by boiling for 6 min. The cooled sample solution was added to a 5% stacking gel and 15% separating gel and subjected to electrophoresis at a constant voltage (80 V). The bands were stained with 0.1% Coomassie blue R-250 for 1 h and decolorized with 7.5% acetic acid and 5% methanol.

### 2.5. Fourier Transform Infrared (FT-IR) Spectroscopy

One milligram of SSP sample was mixed with KBr and pressed into pellets for analysis, which were measured on an FT-IR 3000 spectrophotometer (Jasco Company, Japan) in the range of 4000–400 cm^−1^. The FT-IR data at 1600–1700 cm^−1^ were used for the calculation of the secondary structure of the samples using Origin Pro 8.5 software.

### 2.6. Morphological Analysis

0.1 g of SSP powder was weighed and ground for gold spray treatment (20 s 10 mA) and the elements of the SSP were observed by using Scanning electron microscopy with a scanning electron microscopy-energy dispersive spectrometry (AMETEK, San Diego, CA, USA).

### 2.7. Differential Scanning Calorimetry (DSC) Analysis

The thermal properties of the samples were analyzed using a differential scanning calorimeter (DSC1, Mettler Toledo, Greenville, SC, USA). The samples (5–10 mg) were placed in an aluminum pan, immediately covered with a perforated aluminum lid and heated from 10 °C to 125 °C at a heating rate of 10 °C/min to observe the change.

### 2.8. Se Speciation by High-Performance Liquid Chromatography-Mass Spectrometry (HPLC-MS)

About 0.5 g of SSP samples were dissolved in 50 mL Tris-HCl (30 mM, pH 8.0) buffer. Then, 15 µL of alcalase (Novozymes, 2.4 L FG, 2.4AU/g) was added, and the solution was incubated at 45 °C for 2 h in a thermostatic oscillator. After the addition of 5 mg of proteinase K (Merck, Darmstadt, Germany), 1245680100, 40 AU/g), the solution was incubated at 37 °C for 2 h and then heated in boiling water for 10 min to inactivate the enzymes. Then, the cooled mixture was centrifuged at 6000 rpm for 20 min, and its supernatant was filtered through a 0.22 mm membrane filter. Separation of the Se species was performed with an Agilent 1290 Infinity HPLC system (Agilent, CA, USA). The injection volume was 2 µL, and the analytical column was a ZORBAX 300 A SB-C18 column (100 mm × 2.1 mm; particle size: 3.5 μm). The mobile phase was composed of solvent A (0.1% formic acid in acetonitrile) and solvent B (0.1% formic acid in water)at a flow rate of 300 µL/min. The gradient program was as follows: 0–2 min, A (5%) + B (95%); 2–4 min, A (from 5% to 90%) + B (from 95% to 10%); 4–7 min, A (90%) + B (10%). Se species identification was carried out with an Agilent 6460 triple quadrupole mass spectrometer (Agilent, CA, USA) in the positive ESI, whose capillary voltage was 3500 V with a source temperature of 300 ℃, and the nebulizer gas (N_2_) was also used as the desolvation gas.

### 2.9. Determination of Free Radical Scavenging Activity

#### 2.9.1. Determination of the DPPH Radical Scavenging Activity

The DPPH radical-scavenging activity in the SSP samples was determined according to a previous method [15].

In brief, 0.008 g of DPPH was weighed, and 200 mL absolute ethanol was added to prepare the DPPH ethanol solution, and then different concentrations (0.10, 0.25, 0.75, 1.25, 1.50, 2.00 mg/mL) of SSP sample solution was prepared. 2 mL of sample solution mixed with 2 mL of DPPH ethanol was shaken evenly and preserved in the dark for 30 min, and Vc and Na_2_SeO_3_ were used as reaction controls. The absorbance was measured at 517 nm and was calculated by the following formula (1):Scavenging activity (%) = [(A_0_ − A_x_)/A_0_] × 100(1)

A_0_ in this formula is the absorbance of the blank group (without samples) and A_x_ is the absorbance of the test samples which is added with the different reaction solution.

#### 2.9.2. Determination of the Hydroxyl Radical Scavenging Activity

The hydroxyl radical scavenging activity in the SSP samples was determined according to a previous method [16].

In brief, different concentrations (0.10, 0.25, 0.75, 1.25, 1.50, 2.00 mg/mL) of SSP solutions were prepared and 0.5 mL of 9 mmol/L salicylic acid-ethanol solutions, 0.5 mL of 9 mmol/L ferrous sulfate solution, 5 mL of 8.8 mmol/L H_2_O_2_ solution and an appropriate amount of deionized water were added to incubation and bathed at 37 °C for 20 min to measure the absorbance. The scavenging activity was calculated according to the following formula (2):Scavenging activity (%) = [1 − (A_x_ − A_x0_)/A_0_] × 100(2)

A_0_ is the absorbance of the control (without samples), A_x_ is the absorbance of the different samples, and A_x0_ is the absorbance of blank control (without H_2_O_2_).

#### 2.9.3. Determination of the Superoxide Radical Scavenging Activity

The superoxide radical scavenging activity was determined according to a previous method [17].

In brief, 1 mL of ultrapure water and 4.5 mL of Tris-HCl buffer solution (0.05 mol/L pH 8.2) were mixed and bathed at 25 °C for 20 min. Then 1 mL sample solution and 0.5 mL of pyrogallol (25 mmol/L) were added rapidly and at 25 °C for 5 min, and finally, 1 mL HCl (8 mol/L) was added. The absorbance was measured at 320 nm and the scavenging activity was calculated according to the following formula (3):Scavenging activity (%) = [1 − (Ax − A_0_)/A_x0_] × 100(3)

A_x0_ is the absorbance of the control (without samples), A_x_ is the absorbance of the sample and A_0_ is the absorbance of blank control (without pyrogallic acid).

### 2.10. Determination of Reducing Power

The reducing power of the SSP samples was determined by reducing K_3_[Fe(CN)_6_] according to a previous method [18]. In brief, 2 mL of the sample solution at various concentrations (SSP, Vc or Sodium selenite) was added to 25 mL of PBS buffer (pH 6.6) and 2.5 mL of 1% potassium ferricyanide solution, and the solution was incubated at 50 °C for 30 min. After cooling to room temperature, 1 mL of 10% trichloroacetic acid was added, and the solution was centrifuged at 5000× *g* for 10 min. Then, 2.5 mL of the upper layer was added to 0.5 mL of 0.1% ferric chloride and 2.5 mL of distilled water. The solution was mixed thoroughly and allowed to stand for 10 min. The absorbance was measured at 700 nm.

### 2.11. Ferric Thiocyanate (FTC) Test

To evaluate the antioxidant effect of the samples in the linoleic acid system, the FTC test was performed according to a previous method [19].

In brief, 2 mL of sample solution at various concentrations (SSP, Vc, or Sodium selenite) was mixed with 2.5 mL of 2.5% linoleic acid anhydrous ethanol solution, 4 mL of phosphate buffer (pH = 7), 2 mL of distilled water and the mixture was incubated at 40 °C in the dark and the absorbance was measured every 24 h. Then, 0.1 mL of the mixture was added to 9.7 mL of 75% ethanol and 0.1 mL of 30% ammonium thiocyanate. After incubation for 3 min, 0.1 mL of 0.02 M ferrous chloride in 3.5% HCl solution was added to the reaction solution, and the absorbance was measured at 500 nm.

### 2.12. Statistical Analysis

The results are expressed as the means±S.D.s of triplicate (*n* = 3). Analysis of variance (ANOVA) and Duncan’s multiple range test was used to determine whether the data were significantly different between samples (*p* < 0.05).

## 3. Results and Discussion

### 3.1. Total Se and Protein Se Contents in Sweet Potato Stems

The application of sodium selenite has a certain growth promotion effect on sweet potatoes [20]. High concentrations of Se can cause abnormal growth in spinachand affect its biomass by slowing the growth of roots and reducing the chlorophyll content [21].

As shown in Figure 1A,B, the total Se content in the stems from the soil application treatment increased in a dose-dependent manner, and the total Se content in the stems from the foliar application treatment increased first and then decreased, which is consistent with the results for flue-cured tobacco [22]. However, the total Se content obtained by foliar enrichment was always higher than that obtained by soil enrichment. For SSP, the Se content reached 5.0 μg/g when the soil-applied Se concentration was 10.0 mg/mL, while the protein Se content in the foliar-applied samples was much higher (8.4–38.3 μg/g). To simplify the subsequent experiments, only three SSP samples were selected for the following analyses: SSP (without Se enrichment and with a Se content of 0.14 µg/g), low-Se SSP (foliar application of 0.4 mg/mL Se with a Se content of 8.41 µg/g) and high-Se SSP (foliar application of 2.0 mg/mL Se with a Se content of 38.35 µg/g). Niu et al. (2020) studied the influence of different processing ways on the Se distribution on the tartary buckwheat (0.5677 mg/kg natural Se in soil) and reported the total Se contents existing in Se-enriched buckwheat groat, milled flour, and extruded flake was 118.04 μg/g, 110.55 μg/g and 98.55 μg/g respectively [23]. Betül et al. (2022) reported thathydroponic leeks (selenite or selenite in hydroponic media) had the potential to accumulate Se above 1000 mg/kg without any growth disturbance and noted that selenite was more effectively absorbed by the leeks [24]. Different application methods and different parts of the plant had a certain impact on the distribution of Se, which still needs further study.

### 3.2. Molecular Weight Distribution of SSPs (SDS-PAGE)

The electrophoresis results for the SSPs are shown in Figure 2A. The major molecular weights in the SSPs (without β-ME) were 15.0, 22.0, 30.0, and 31.6 kDa, among which the bands at 22.0 and 31.6 kDa are similar to those of sporamin A (22 kDa) and sporamin B (31 kDa), the major storage protein in sweet potato tubers [25]. In comparing the electrophoresis patterns of the SSP, low-Se SSP and high-Se SSP, the electrophoresis bands appeared shallow, and the major molecular weights increased after Se enrichment; this may have occurred because sulfur (S) was replaced by Se in proteins and made their molecular weights slightly higher due to their altered structures [26].

### 3.3. FT-IR Analysis

As shown in Figure 2B, the SSPs showed typical amide vibrations at 1220–1310 cm^−1^ (amide III, C-N and C-O stretching, O = C-N and N-H bending), 1460–1560 cm^−1^ (amide II, N-H bending and C−N stretching), 1610–1680 cm^−1^ (amide I, C=O stretching) and 3310–3350 cm^−1^ (amide A, N-H stretching) [27]. However, obvious differences were observed between Se-enriched and non-Se-enriched SSPs. When the wavenumber was below 800 cm^−1^, there was no obvious absorption peak in the spectrum of the SSP, while absorption peaks at 551 cm^−1^ and 612 cm^−1^ in the high-Se SSP were attributed to Se-C and Se-O-C, respectively [28]. In addition, absorption peaks at 634 cm^−1^ and 2260 cm^−1^ were observed in the low-Se SSP, which was associated with Se = C and Se-H, respectively [29]. The peak at 1045 cm^−1^ in the Se SSPs was attributed to O-Se-O asymmetric tension [30]. From the results above, there were significant differences in Se bonds among the three samples and the number of forms of Se chemical bonds gradually increased with increasing Se content, such as the Se-O bond in high Se-enriched SSP, which could be concluded that Se is successfully incorporated into the proteins in the growth process of Se-enriched sweet potato instead of simply being wrapped in the protein structure.

Through the calculation of the FT-IR data, the secondary structure compositions of the SSPs were obtained (Table 1). Compared with those of the SSP, the percentages of β-turns increased and the percentages of α-helices and random coils decreased in the Se-enriched samples. Compared with those in the low-Se SSP, the percentages of α-helices, β-turns and random coils in the high-Se SSP increased; however, the percentages of β-sheets decreased from 54.5% to 34.8%, which may have been caused by the decrease in SeCys_2_. According to previous studies, C-Se may exist as methylselenocysteine (MeSeCys), selenomethionine (SeMet), or selenocysteine (SeCys_2_), and Se may convert C-S-S-C (SeCys_2_) into C-Se-Se-C, which means that Se can influence the percentage of β-sheets by promoting the transformation of SeCys_2_ [31].

### 3.4. SEM-EDS Analysis

The results for the element compositions and surface characteristics are shown in Table 2 and Appendix A respectively. The surface of the SSPs remained intact, smooth and tight, and there were no obvious differences among the samples. However, major differences in the contents of the main elements were observed among the samples. With the increase in Se applied, the Se atomic intensity of the sample surface gradually increased (SSP 0.04%, low-Se SSP 0.06% and high-Se SSP 0.09% in atomic percentage, At%). In addition, the carbon content (atomic percentage, At%) decreased from 71.9 to 58.8, and the oxygen content increased from 16.1 to 25.7 with the increase in Se content in the SSPs. This finding suggests that the incorporation of Se into the proteins mainly occurs through covalent binding, such as C- and O-containing functional groups, and the results above are consistent with those of the FT-IR analysis in Figure 2B.

### 3.5. DSC Analysis

As shown in Figure 2C, SSPs showed relatively obvious endothermic peaks at approximately 25 °C, and as the Se concentration increased, the decomposition endothermic peaks moved to slightly higher temperatures at first and then moved back slightly. Compared with those of the SSP, the α-helix contents of Se-enriched SSPs decreased significantly in Table 1 and this decrease indicates that the proteins were partially unfolded, which was related to the breaking of hydrogen bonds between or within the molecules of the proteins [32]. Furthermore, compared with those of the low-Se SSP, the contents of β-turn and the random coil of high-Se SSP were increased, and sometimes the increased β-turn may be transformed from other more ordered structural units [33,34]. The content of random coils in high-Se SSP increased compared with the low-Se SSP, which suggests increased flexibility and decreased stability of protein [35,36,37].

### 3.6. Se Speciation in SSPs

HPLC-MS was used to analyze the Se species in the samples after enzymolysis. According to previous reports, MeSeCys was regarded as the main seleno-amino acid after Se enrichment in Se-enriched leeks and mung bean sprouts [24,38]. Tie et al. reported that MeSeCys was the major organic Se form in mung bean sprouts, which reached a maximum of 1817.8 μg/kg, accounting for 42% of the total Se content [38]. The chromatograms of the low-Se SSP and high-Se SSP after enzymatic hydrolysis and the contents of Se species are shown in Figure 3A,B, respectively. As shown in Figure 4B, no inorganic Se species were detected in the SSP samples, and MeSeCys was the major form of Se in the low-Se SSP (15.2 mg/g), followed by SeMet (3.0 mg/g). However, with the increase in Se content, the content of MeSeCys (1.9 mg/g) was greatly reduced in the high-Se SSP, but the content of SeMet (3.4 mg/g) increased slightly. In addition, the SeCys_2_ content decreased with increasing Se content in the samples. The results showed that the concentration of Se species does not increase in proportion to Se enrichment, and SeMet is the primary Se species in the high-Se SSP, as was found in grain and forage crops [39].

### 3.7. Free Radical Scavenging Activities of SSPs

#### 3.7.1. DPPH Radical Scavenging Activity

The DPPH free radical scavenging assay is currently an effective method for assessing the ability of a sample to remove free radicals. As shown in Figure 4A, the DPPH radical scavenging activity of the samples increased with the sample concentration, but when the concentration was higher than 0.75 mg/mL, the increase in the scavenging activity tended to be slight. Of the samples, inorganic Se showed the lowest DPPH radical scavenging activity, and ascorbic acid (Vc) exhibited the highest scavenging activity for DPPH free radicals. The Se-enriched SSPs showed significantly higher DPPH scavenging activity. The scavenging activity of the high-Se SSP was slightly higher than that of the low-Se SSP, but the difference was not significant. The peak DPPH scavenging activities of Vc, high-Se SSP, low-Se SSP and SSP were 95.5%, 71.6%, 70.6% and 66.2%, respectively.

#### 3.7.2. Hydroxyl Radical Scavenging Activity

As shown in Figure 4B, the SSPs exhibited a certain ability to scavenge hydroxyl radicals. The scavenging activity increased with increasing sample concentration, but the scavenging activity was generally low. The scavenging activity of SSPs was much lower than that of Vc and was also lower than that of inorganic Se. The high-Se SSP showed higher hydroxyl radical scavenging activity than the low-Se SSP and SSP.

#### 3.7.3. Superoxide Anion Radical Scavenging Activity

Compared with inorganic Se, the SSPs showed higher superoxide anion radical scavenging activity, and the scavenging activity increased significantly as the sample concentration increased in Figure 4C. When the Se concentration was in the range of 0.1 to 0.25 mg/L, the scavenging activity of the low-Se SSP was higher than those of SSP and high-Se SSP. However, in the range of 0.25–0.75 mg/mL, the high-Se SSP exhibited a large increase in scavenging activity. In addition, the highest scavenging effect of SSPs (96.7%) was slightly lower than that of Vc (97.1%) at the concentration of 2 mg/L.

The antioxidant capacities of Se-enriched proteins extracted from Se-enriched products, such as green tea, rice, and garlic were investigated, and there was also a significant positive correlation between Se content and antioxidant capacity [40]. Furthermore, the proteins extracted from Se-enriched peanut leaves and mushrooms also showed better radical scavenging activities against hydroxyl radicals and superoxide anion radicals compared to the proteins without Se-enrichment [13,14]. Se-enriched SSPs demonstrated potent scavenging activities against DPPH radicals and superoxide anion radicals. However, the Se SSPs showed lower hydroxyl radical scavenging activity, which might be due to the different free radical generation mechanisms.

### 3.8. Reducing Power and Ferric Thiocyanate (FTC) Test for SSPs

As shown in Figure 5A, the reducing power of inorganic Se was very low, and the reducing power of Vc was very strong. The reducing power of the SSP samples increased with the increasing Se concentration, and the reducing power of the SSPs did not differ significantly in the concentration range of 0.1 to 0.75 mg/L. However, the reducing power of the high-Se SSP was significantly higher than those of the low-Se SSP and the SSP in the concentration range of 0.75–2 mg/mL.

The more severe the linoleic acid oxidation, the higher the absorbance value, which indicates that the antioxidant capacity of the reagent is weak. As shown in Figure 5B, the high-Se SSP had a stronger antioxidant capacity than Vc, which was related to the fact that Vc has poor solubility in the linoleic acid autoxidation system. The antioxidant capacity of the high-Se SSP was high within a certain Se concentration range.

## 4. Conclusions

The Se content in sweet potato stems was effectively increased by foliar Se application. The proteins were extracted from the stems, and their molecular weights were determined by SDS-PAGE. The result of FT-IR showed that there were significant differences in Se bonds among the samples at different concentrations and the number of forms of Se chemical bonds gradually increased with increasing Se content, such as the Se-O bond in high Se-enriched SSP, it can be concluded that Se is successfully incorporated into the proteins in the growth process of Se-enriched sweet potato instead of simply being wrapped in the protein structure. The changes in the secondary structures in SSPs demonstrated that the increase in β-turns and random coils improved the thermal stability of the Se-enriched SSPs. MeSeCys, SeCys_2_, and SeMet were observed in the Se SSPs, and MeSeCys was the main component in the low-Se SSP. Se-enriched SSPs showed potent DPPH radical and superoxide anion radical scavenging activities, and the antioxidant capacities were related to the Se content to some extent, indicating that Se-enriched SSPs can be used as dietary Se supplements or as antioxidants.

## Figures and Tables

**Figure 1 foods-10-03064-f001:**
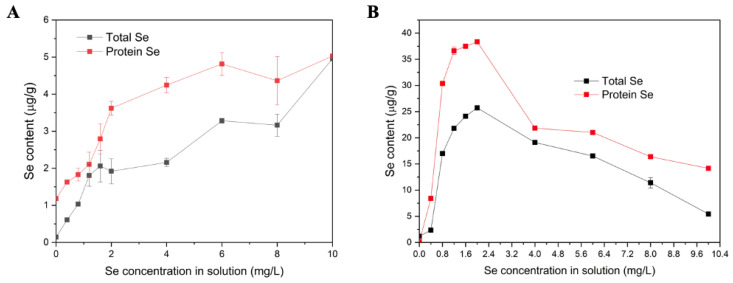
Se contents in SSPs and sweet potato stem powder cultivated by different Se enrichment methods: (**A**) soil application, (**B**) foliar application. Values are means ± SDs (*n* = 3).

**Figure 2 foods-10-03064-f002:**
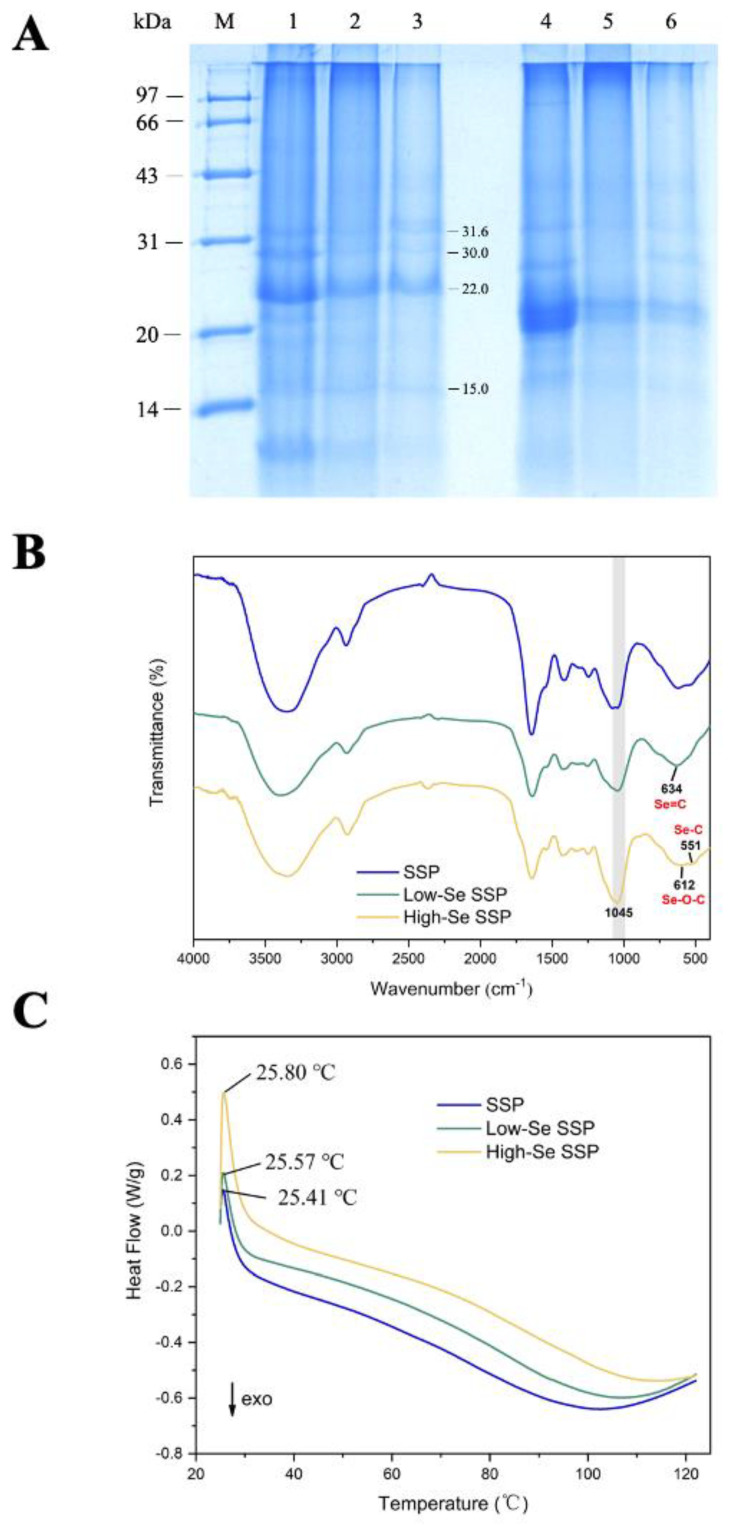
SDS-PAGE analysis (**A**) of SSP (1), low-Se SSP (2) and high-Se SSP (3) under nonreducing conditions and SSP (4), low-Se SSP (5), and high-Se SSP (6) under reducing conditions. M: standard protein markers; FT-IR spectra of SSP samples (**B**); DSC curves of SSP samples (**C**).

**Figure 3 foods-10-03064-f003:**
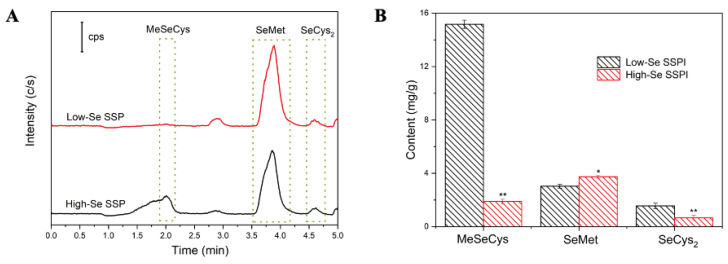
(**A**) HPLC-MS chromatograms of SSP samples. (**B**) Se species compositions in SSP samples. Values are means ± SDs (*n* = 3). * represent *p* < 0.05 and ** represent *p* < 0.01.

**Figure 4 foods-10-03064-f004:**
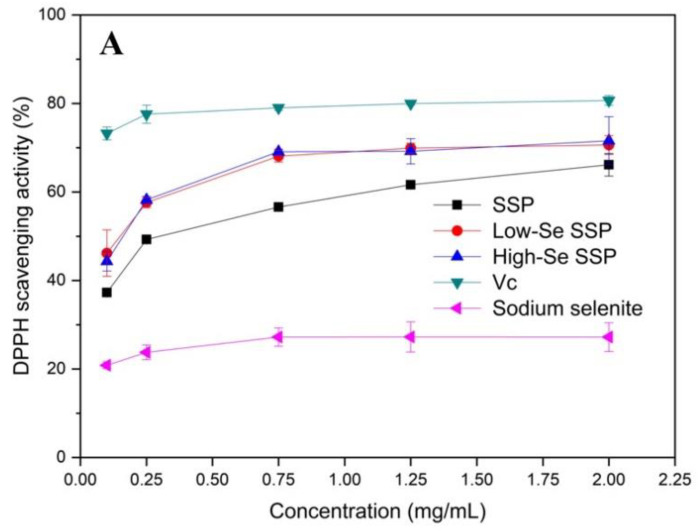
Scavenging activities of SSP, low-Se SSP, high-Se SSP, Vc (positive control)and sodium selenite (inorganic Se) at various concentrations against 1,1-diphenyl-2-picrylhydrazyl radicals (**A**), hydroxyl radicals (**B**) and superoxide radicals (**C**). Values are means ± SDs (*n* = 3).

**Figure 5 foods-10-03064-f005:**
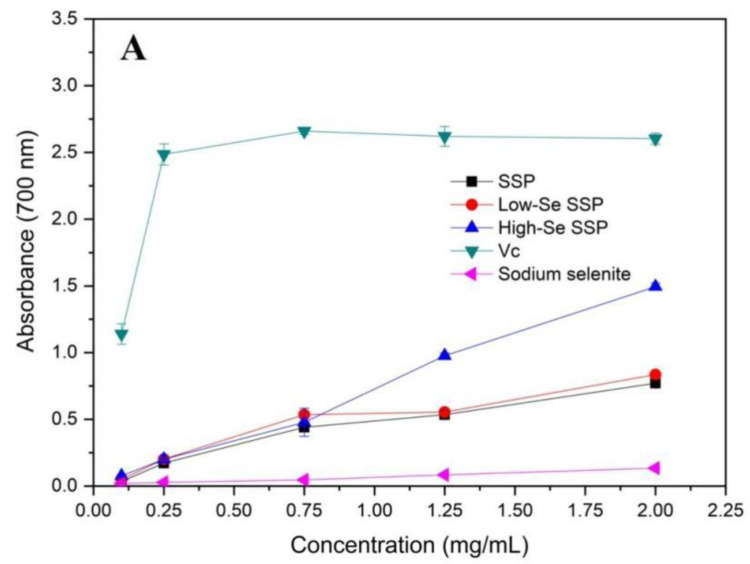
Reducing power (**A**) and ferric thiocyanate (FTC) test (**B**) of SSP, low-Se SSP, high-Se SSP, Vc (positive control), and sodium selenite (inorganic Se) at different concentrations. Values are means ± SDs (*n* = 3).

**Table 1 foods-10-03064-t001:** Estimation of the secondary structural composition of SSP, low-Se SSP, and high-Se SSP by FT-IR spectra.

Sample Composition of the Secondary Structure (%)				
	α-Helixes	β-Sheets	β-Turns	Random Coils
SSP	31.94 ± 1.90 a	15.63 ± 0.25 c	32.52 ± 0.95 a	19.90 ± 0.91 a
Low-Se SSP	8.34 ± 0.15 c	54.53 ± 0.35 a	35.49 ± 0.15 b	1.74 ± 0.16 c
High-Se SSP	15.00 ± 0.18 b	34.8 ± 0.72 b	36.80 ± 1.07 b	13.40 ± 1.9 b

Results (mean ± SD, *n* = 3) with different letters with the same column are significantly different at *p* < 0.05.

**Table 2 foods-10-03064-t002:** Weight and atomic percentages of elements in SSP, low-Se SSP, and high-Se SSP as determined by SEM-EDS.

Element	SSP		Low-Se SSP		High-Se SSP	
	at%	±SD	at%	±SD	at%	±SD
C K	72.00	4.16 a	61.00	5.73 b	58.80	5.11 b
O K	16.10	1.06 b	23.10	2.42 a	25.70	1.55 a
N K	2.78	0.40 b	11.80	1.97 a	10.80	1.60 a
P K	0.87	0.21 a	0.56	0.11 ab	0.51	0.10 b
S K	0.87	0.11 a	0.73	0.11 a	0.41	0.06 b
Se K	0.04	0.02a	0.06	0.02a	0.09	0.04b
Na K	3.46	0.89a	2.09	0.23c	3.00	0.26ab
Cl K	3.92	0.40a	0.75	0.13b	0.72	0.05b

at% represents atomic percentage and results (mean ± SD, *n* = 3) with different letters in the same row are significantly different at *p* < 0.05.

## Data Availability

The datasets generated for this study are available on request to the corresponding author.

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
