# Peer review of "Se-O Bond Is Unique to High Se Enriched Sweet Potato Stem Protein with Better Antioxidant Ability"

_foods, 2021, doi:10.3390/foods10123064_

Round 1
Reviewer 1 Report
Very interesting theme and very good concept of study. I can suggest major revision for the paper entitled “Se-O bond is unique to high Se enriched sweet potato stem protein with better antioxidant ability”. This is my suggestion for improvement of this paper:
Abstract has to have a little promising moment in the end. Please add why your results are good, further perspective(s), etc.
2.4. Sodium dodecyl sulfate-polyacrylamide gel electrophoresis – although the Author wrote that method was described in some previously papers, it is necessary to emphasize the main parts of the method
2.6. Morphological analysis – please provide information about the preparation of samples, gold covering, etc.
For subsection 2.9., 2.10. and 2.11. is the same comment as for 2.4.
2.12. Statistical analysis – the letter of this subsection is bigger than others
Presenting the results of this study is good, but can be improved. For example, the Authors need to remark on the importance of selenium. For example, emphasize that selenium is a naturally occurring mineral required for good health. It is obtained from food, and the recommended dietary allowance is 55 μg/d for persons 14 years or older, with a tolerable upper intake limit of 400 μg/d. etc, etc. After that, all subsections need more comparisons with other papers which deal with the same topic (selenium-enriched food or raw materials).
The presented figures are not clear. Please provide a better quality of the pictures.
Section Conclusions required further perspective(s) of this study.
Suggestion for Authors: You can really upgrade your investigation using advanced mathematical models for predictions. Statistical methods used in this paper are the biggest weakness of this study, but you could have an elegant solution for better research presenting the obtained results.
Reviewer 2 Report
The paper fulfils the aims and scope of Foods. The article ‘Se-O bond is unique to high Se enriched sweet potato stem protein with better antioxidant ability’ presented for review is interesting. However, some issues need to be clarified or supplemented. The comments are included below.
Title
The title is worded correctly and accurately reflects the content.
Abstract
The abstract is clear and adequate.
Materials and methods
2.2. Extraction of SSP
- Line 82-84: Why was such a long spin time? Was the sample cooled during centrifugation?
2.8. Se speciation by high-performance liquid chromatography-mass spectrometry (HPLC-MS)
- At what temperature was the analysis carried out? Which gradient was used? The information provided should allow for the analysis by the reader of the article.
Results
The results are well-discussed. However, there are some issues that require clarification.
- Where are the results of the statistical analysis posted?
- Is it better not to mark the figure in the attachment with a different number if the text already contains figure 1. This remark is debatable and please comply with the requirements of the journal.Conclusions
- The conclusions section is correct.
Others
The literature is correct.
Reviewer 3 Report
Review of the research article: "Se–O bond is unique to high Se enriched sweet potato stem protein with better antioxidant ability
Enriching plants with important and often deficient elements for humans and animals, such as selenium or iodine, is a promising field of science. Sweet potato, as a vegetable grown on a large area, in many countries, is certainly a valuable object of research here. The Authors enriched the sweet patato stems with selenium. However, they did not mention how, in their opinion, the stems could be used. They wrote only that they are discarded for now. Are they sure the stems are free from poisonous compounds? The text could contain 2–3 sentences about future perspectives.
I do not quite understand the title of the work. The Se–O bond appears four times in the work: in the title, in the abstract, and twice in the section. 3.3. "FT-IR analysis". It is not even mentioned in conclusions. It does not appear anywhere as the main thesis of the work. Either I would complete the text or I would change the title.
The keywords are partially wrongly chosen. The symbol Se appears in the title. So there is no need to repeat it among keywords. Here the Authors would simply use 'selenium' (which appears in its full form only twice in the manuscript). The keyword 'amino acid' seems to be too general. Instead, I would choose a word that is more directly related to this article.
The advantages of the work are understandable language and legible figures.
My other remarks mainly concern the methodology, which in my opinion is not carefully presented and is the weakest element of the article.
In the section 2.2. "Extraction of SSP: the Authors started with:"A certain amount of sweet potato stem powder ..." and ended: "The SSP solution was lyophilized and stored..." (with no amount). Such a general description means that we are not able to imagine the scale or efficiency of the extraction. At least approximate data is needed here.
In the section 2.4. "Sodium dodecyl sulfate-polyacrylamide gel electrophoresis (SDS-PAGE) of SSP" the Authors refered to the methodology described in [14]. Unfortunately, this is not an open access article (it is not available at my research unit).
Similarly, in the section 2.10. "Determination of reducing power", the Authors refered to the work [18]. Unfortunately, this is also not an open access peper (access costs 42.99 Euro). If the Authors chose non-open access references, they should include at least the method name. There are several methods for determining the reducing potential (e.g. FRAP, CuPRAC). Nevertheless, it would be elegant to refer to the methodology that is generally available if the Authors resign from its description.
Moreover, if the Authors applied the method used by other authors to other plants, they made some modifications with high probability. Leaving methodological differences not described in the text gives the impression of a lack of diligence.
Minor remarks:
The Authors specified the spectrofluorimeter as a "FT-IR spectrophotometer (Jusco Company, Japan)" Is this a correct name? Jasco (from Japan) is a well-known big company that produces spectrofotometers.
In the section 2.12. "Statistical analysis" the Authors wrote: "The results are expressed as the means ± S.D.s of three parallel measurements of each sample". I understood that each sample was measured in triplicate. However, the Authors forgot to write how many samples they made for analysis. This can be found out by analyzing the "Figure captions" at the bottom of the manuscript. Whole the section 2.12 was written with a bigger font than the whole text.
I also do not understand why the symbol and origin of some equipment (such as an ultrasonic cleaner) were described in great detail, while those mentioned in the sentence below (centrifuge) were not at all (please, compare in lines 81 and 82). I would decide to standardize, in terms of complexity, the way of describing the equipment.
The Authors omitted the explanation that 'beta-ME' means beta-mercaptoethanol.
Nov. 14, 2021
Round 2
Reviewer 1 Report
Dear Editor,
After checking revised paper, I can suggest acceptance of this manuscript.
Best regards,
Reviewer 3 Report
The manuscript ID foods-1450811 has been corrected and completed. I have no further comments or questions.
Nov. 30th 2021